# Fabrication of Large-Area Mullite–Cordierite Composite Substrates for Semiconductor Probe Cards and Enhancement of Their Reliability

**DOI:** 10.3390/ma15124283

**Published:** 2022-06-17

**Authors:** Da-Eun Hyun, Jwa-Bin Jeon, Yeon-Sook Lee, Yong-Nam Kim, Minkyung Kim, Seunghoon Ko, Sang-Mo Koo, Weon Ho Shin, Chulhwan Park, Dong-Won Lee, Jong-Min Oh

**Affiliations:** 1Department of Electronic Materials Engineering, Kwangwoon University, 20 Kwangwoon-ro, Nowon-gu, Seoul 01897, Korea; hde1704@ktl.re.kr (D.-E.H.); hdjbj3@kw.ac.kr (J.-B.J.); minkyungkim@kw.ac.kr (M.K.); shko@kw.ac.kr (S.K.); smkoo@kw.ac.kr (S.-M.K.); weonho@kw.ac.kr (W.H.S.); 2Material Technology Center, Korea Testing Laboratory, 87 Digital-ro 26-gil, Guro-gu, Seoul 08389, Korea; lou0824@ktl.re.kr (Y.-S.L.); ynkim@ktl.re.kr (Y.-N.K.); 3Department of Chemical Engineering, Kwangwoon University, 20 Kwangwoon-ro, Nowon-gu, Seoul 01897, Korea; chpark@kw.ac.kr

**Keywords:** mullite, cordierite, composite ceramics, environmental tests, semiconductor probe cards

## Abstract

This work aims to fabricate a large-area ceramic substrate for the application of probe cards. Mullite (M) and cordierite (C), which both have a low thermal expansion coefficient, excellent resistance to thermal shock, and high durability, were selected as starting powders. The mullite–cordierite composites were produced through different composition ratios of starting powders (M:C = 100:0, M:C = 90:10, M:C = 70:30, M:C = 50:50, M:C = 30:70, and M:C = 0:100). The effects of composition ratio and sintering temperature on the density, porosity, thermal expansion coefficient, and flexural strength of the mullite–cordierite composite pellets were investigated. The results showed that the mullite–cordierite composite pellet containing 70 wt% mullite and 30 wt% cordierite sintered at 1350 °C performed exceptionally well. Based on these findings, a large-area mullite–cordierite composite substrate with a diameter of 320 mm for use in semiconductor probe cards was successfully fabricated. Additionally, the changes in sheet resistance and flexural strength were measured to determine the effect of the environmental tests on the large-area substrate such as damp heat and thermal shock. The results indicated that the mullite–cordierite composite substrate was extremely reliable and durable.

## 1. Introduction

The probe card is a device that is used to verify the operation of the integrated circuit chip on the wafer. Probe tips, a space transformer, interposers, and a printed circuit board (PCB) comprise the probe card. The density of probe cards has also increased significantly in recent years, owing to the high degree of integration of semiconductor devices [1]. While increasing the number of wirings on the PCB increases the density of probe cards, the main PCB becomes difficult to accommodate the rapidly increasing wiring [2]. To address this issue, the majority of the probe cards accommodate a portion of the wiring by sandwiching a ceramic substrate between the contact probe and the PCB as a space transformer. A space transformer’s ceramic substrate must meet a number of requirements. To begin, the probe card’s thermal expansion coefficients (TEC) must be comparable to those of silicon [3,4]. When the silicon wafer is heated to 150 °C for wafer-level burn-in, the probe card cannot be used due to dimension errors between the probe pins and the pads caused by the probe card’s thermal expansion mismatch with the silicon wafer [4]. Additionally, as the wafer’s size has increased in recent years, it is susceptible to temperature changes. As a result of this thermal expansion, the space transformer’s position of the probe pin changes. Consequently, a TEC comparable to that of the silicon wafer is required. Second, it is critical to control the sintering shrinkage of the probe card’s ceramic substrate due to the high number of probe pins. Due to the shrinkage of the ceramic substrate during the sintering process, the positions of the processed holes may deviate from the designed positions. As a result, it is critical to consider ceramics with a low TEC as potential substrate materials for semiconductor probe cards and to investigate the various factors affecting ceramic sintering.

Cordierite (2MgO·2Al_2_O_3_·5SiO_2_) is a promising material for the probe card substrate due to its attractive properties, which include high thermal shock resistance, high resistivity, high refractoriness, chemical stability, and low TEC. As a result, cordierite has found widespread use in a variety of applications, including electrical insulators, refractories, filters, membranes, integrated circuit board substrates, and microwaves [5,6,7,8]. However, dense cordierite ceramics are difficult to obtain due to cordierites’ narrow sintering temperature and difficulty sintering using the solid-state process [9,10,11].

Numerous studies have examined various methods for increasing the densification of cordierite ceramics through the addition of sintering aids. Banjuraizah et al. [12] discovered that adding magnesium oxide to cordierite glass improves densification, but that adding too much magnesium oxide increases the TEC. Chen [13,14] also increased the density of glass-ceramics based on cordierite by incorporating sintering aids such as CaO and ZnO. However, excessive use of these sintering aids increased the TEC and dielectric loss. As a result, it is critical to choose appropriate additive materials in order to fabricate a dense ceramic substrate with a low TEC for use in probe cards.

Mullite (3Al_2_O_3_·2SiO_2_) is well-known for its ability to enhance the mechanical properties of cordierite [7,15,16], but it is also used in electronic substrates, microelectronic packaging, and as a component in reinforced composites due to its high electrical resistance, thermal shock resistance, excellent thermal and chemical stability at elevated temperatures, and good dielectric and mechanical properties [16,17,18,19]. Albhilil et al. [10] reported that although the TEC increases slightly with the addition of mullite, the relatively low mechanical strength of cordierite can be improved through polymorphism transformation. Subramanian et al. [20] proposed that by matching the TEC of the probe card substrate to that of a silicon wafer, chip detachment and device failure can be avoided. Additionally, they reported that a mullite–cordierite composite with a volume ratio of 35:65 has a TEC that matches Si.

The composition ratio of composite ceramics and the factors affecting sintering were investigated in order to fabricate mullite–cordierite composite substrates with superior physical and mechanical properties. This resulted in the fabrication of a large-area mullite–cordierite composite substrate for testing IC chips formed on the most commonly used 12-inch wafer. Additionally, the sheet resistance and flexural strength of the large-area mullite–cordierite composite substrate were measured during various environmental tests to determine its reliability. 

## 2. Materials and Methods

### 2.1. Materials and Processing

As starting powders, commercially available mullite (DAIHAN REFRACTORIES MATERIAL, Seoul, Korea) and cordierite (Eastking Industrial Limited, Luoyang, China) were labeled M and C, respectively. Mullite and cordierite powders had median diameters (*D*_50_) of 3.96 µm and 4.23 µm, respectively. The starting powders were then subjected to 24 h of attrition milling to determine the effect of particle size and size distribution on granule formation. Following that, a slurry was prepared to produce granules through different composition ratios of starting powders (M:C = 100:0, M:C = 90:10, M:C = 70:30, M:C = 50:50, M:C = 30:70, and M:C = 0:100). To this slurry, polyvinyl alcohol (PVA) used as binder, dispersant, and releasing agent was added. Sintering aids such as TiO_2_, Y_2_O_3_, Cr_2_O_3_, Co_2_O_3,_ and MnO powders were then added to the slurry. The slurry was then ball-milled for 3 h with ZrO_2_ balls to achieve a uniform dispersion and spray dried. After spray drying, the dried granules were sieved and then uniaxially pressed into cylindrical pellets (36 mm in diameter and 6 mm in thickness) at an applied pressure of 80 MPa. Then, these pellets were dried at 400 °C for 20 h and sintered for 4 h at a temperature range of 1300 °C to 1450 °C. To determine the optimal mixing ratio and sintering temperature for the mullite–cordierite composites, the physical and mechanical properties of the mullite–cordierite composites were analyzed. Figure 1 illustrates the processing flow for mullite–cordierite composites.

### 2.2. Fabrication of Large-Area Ceramic Composite Substrate

To fabricate a large-area ceramic composite substrate for the semiconductor probe card, the granules of the mullite–cordierite composite with the optimal composition ratio were uniaxially pressed at 80 MPa into 320 mm diameter ceramic substrates. Then, this ceramic composite substrate was dried at 400 °C for 20 h and sintered for 4 h at 1350 °C. Following that, the ceramic substrates were subjected to surface grinding since the substrate’s shrinkage and warpage may have occurred during the sintering process. Ultrasonic drilling was performed, and 30,000 micro-holes were processed through it, taking into account the frequency and amplitude of the ultrasonic generator, the size of the processing pin, and the processing pressure. The details of the ultrasonic drilling process were as follows: (1) The frequency and amplitude of the ultrasonic generator: 20 kHz and 20 µm, respectively. (2) The size of the processing pin: 0.4 mm of piano wire. (3) The processing pressure was controlled differently in each step to prevent chipping of the processing product: 0.3 g/pin in the initial stage, 2 g/pin in the middle stage, and 0.5 g/pin in the final stage.

### 2.3. Characterization Procedure

The average particle size of the starting powders was performed by a particle size analyzer (PSA, Bluewave, MICROTRAC, Osaka, Japan), which is a wet measuring instrument based on the laser diffraction principle. Field-emission scanning electron microscopy (FE-SEM, MIRA3 XMU, TESCAN, Brno, Czech Republic) was used to examine the microstructure of the mullite–cordierite composites. The bulk density and porosity were determined using the static weighing method according to ASTM C20 [21]. The thermomechanical analyzer (TMA, Q400, TA Instrument, New Castle, DE, USA) was used to determine the TEC of the mullite–cordierite composites over a temperature range of 25–500 °C at a heating rate of 5 °C/min.

To determine the durability and reliability of ceramic substrates made of mullite–cordierite composites, 80 mm × 80 mm specimens were processed from the fabricated ceramic substrates. Two environmental tests were performed to ascertain the specimens’ degradation characteristics under harsh environmental conditions. A damp heat (DH) test was conducted in a chamber (WKE 64/70, WEISS, Monroe, NC, USA) at 85 °C/85% for 100 h. The thermal shock (TS) test was conducted in a thermal shock chamber (TSA-41L, ESPEC, Osaka, Japan) for 100 cycles. One cycle consists of heating the substrate to 85 °C for 30 min and then cooling it to −40 °C for 30 min. 

Following the environmental tests, the ceramic substrate’s sheet resistivity and flexural strength were determined. The resistivity of the sheet was determined at 1000 V using a high resistance meter (Hiresta-UX MCP-HT 800, MITSUBISHI CHEMICAL ANALYTECH, Yamato, Japan). Flexural strength was determined using the universal testing machine via a three-point flexural test (UTM, INSTRON, NVLAP, USA). Rectangular bars with dimensions of 3 mm × 4 mm × 36 mm were prepared. During the flexural test, the specimens were subjected to a constant flexural strength until failure occurred, with the crosshead speed set to 0.5 mm/min. The three-point flexural strength ($\sigma_{f}$) was calculated according to the following formula:$$\sigma_{f}=\frac{3FL}{2bd^{2}},$$
where *F* is the load, *L* is the length of the span, *b* is the width of the specimen, and *d* is the specimen thickness. The flexural strength data was calculated using an average of ten specimen measurements.

## 3. Results and Discussion

### 3.1. Formation of Mullite–Cordierite Composites Granules by Spray Drying Process

Sintering is used to produce mullite–cordierite composites with high mechanical strengths, low TEC, and dense structures. Numerous variables affect the properties of the sintered body during the sintering process, including the particle size distribution and morphology of the powders, the sintering temperature, and sintering aids. The particle size distribution of the powders has a significant effect on both the reaction rate and sintering behavior. For instance, in coarse powders with a low initial density due to non-uniform particle size and high agglomeration, the particles are separated by a large distance, resulting in a slow rate of densification. As a result, it is critical to maintain control over the particle size distribution of the starting powders.

Figure 2 shows the particle size distributions of the starting powders before and after attrition milling. As shown in Figure 2a, for the mullite powder, the peak of the particle size distribution curve (which has a unimodal distribution) moved toward the smaller particle size after attrition milling, and the *D*_50_ value decreased from 3.96 to 1.75 µm. As shown in Figure 2b, after attrition milling, the particle size distribution of the cordierite powder shifted from bimodal to unimodal, and the *D*_50_ value decreased from 4.23 to 3.23 µm. The SEM images in Figure 2 demonstrate that the initial powders with a range of particle size transformed into homogeneous powders with uniform and small particle sizes after milling. It was anticipated that starting powders with particle sizes greater than 10 µm would be crushed during the milling process.

Figure 3 shows the microstructure of granules formed by spray drying the mixed powders of mullite and cordierite powders prior to and after attrition milling. It demonstrates the effect of the particle size distribution of mullite and cordierite powders on granule formation. Granules formed prior to attrition milling had a variety of morphologies (Figure 3a). In comparison, granules formed using starting powders with homogeneous size distribution obtained via the attrition milling process were found to be spherical (Figure 3b). Additionally, the un-milled mixed powders had a higher *D*_50_ value (147.3 µm) than the attrition milled mixed powders (88.31 µm). The span of the particle size distribution is defined as (*D*_90_–*D*_10_)/*D*_50_. The span of the un-milled mixed powders was greater. Before attrition milling, the span was 1.22; after attrition milling, the span was 0.71. These findings demonstrate that the particle size and size distribution of the initial powder has an effect on the morphology of spray-dried granular powders. Previously published work demonstrated that granules formed from finer initial powders and powders with a narrow size distribution formed smoother and denser structures than those formed from coarse initial powders. Additionally, the powders’ spherical granules exhibit good flowability, which is critical for the fabrication of dense and mechanically strong ceramics [22]. As a result, the spherical granules obtained after spray drying the fine powders are suitable for fabricating mullite–cordierite composite pellets and substrates.

### 3.2. Characterization of the Sintered Mullite–Cordierite Composite Pellets Produced by Varying the Cordierite Content

Prior to fabricating a large-area composite ceramic substrate, it is critical to determine the optimal composition ratio and sintering temperature for improving the mechanical and physical properties of sintered mullite–cordierite composite pellets. To begin, we investigated the effect of composition ratio on sintered mullite–cordierite composite pellets (36 mm in diameter and 6 mm in thickness). The fractured surfaces of mullite–cordierite composite pellets with various cordierite weight ratios (0, 10, 30, 50, 70, and 100 wt%) sintered at 1300 °C are shown in Figure 4. The sample containing 100 wt% cordierite exhibited a porous structure, whereas, the sample containing 100 wt% mullite exhibited a more compact structure. For samples of pure cordierite and mullite, the density (cordierite = 2.55 g/cm^3^ and mullite = 3.02 g/cm^3^, respectively) was measured to be similar to the average theoretical density (cordierite = 2.53 g/cm^3^ and mullite = 3.17 g/cm^3^, respectively) [23], which is attributed to the structure of the samples, as shown in Figure 4a,f. As the mullite content of the composite pellets increased, the density increased noticeably, as shown in Figure 4c–e, compared to Figure 4b. Thus, increasing the mullite content not only accelerates the densification process during sintering, [24] but also increases the overall density of the samples, as mullite has a higher density than cordierite. Additionally, a previous study reported that in batches with >90 wt% cordierite, the cracks formed by the cristobalite transition from β to α weakened the sample’s structure [23].

The effect of the cordierite content on the TEC and flexural strength of sintered composite pellets is shown in Figure 5. In general, to improve thermal shock resistance, the ceramic substrate for probe cards should have a TEC close to that of Si. As shown in Figure 5a, the TEC of the samples decreased as the cordierite content increased. The sample containing 70 wt% cordierite had the lowest TEC (2.88 ppm/°C) of any sample (except the pure cordierite pellet) and demonstrated excellent thermal shock resistance. However, when compared to the Si TEC (3.2–3.9 ppm/°C) [25], it can be assumed that all composite pellets containing 30–70 wt% cordierite have high thermal stability. As shown in Figure 5b, the flexural strength of samples containing up to 30 wt% cordierite is significantly greater than that of samples containing more than 50 wt% cordierite. This is due to mullite’s greater mechanical strength compared to cordierite. Thus, samples containing a high concentration of mullite have greater flexural strength. Additionally, the figure shows the relationship between density and flexural strength. Flexural strength increases as density increases, as shown in Figure 4 and Figure 5b. As a result of the low TEC and high flexural strength of mullite–cordierite composites, it can be concluded that a composition ratio of 70 wt% mullite and 30 wt% cordierite is optimal for forming mullite–cordierite composites.

### 3.3. Characterization of Mullite–Cordierite Composite Pellets Sintered at Different Temperature

Figure 6 shows the physical and mechanical properties, such as density, porosity, flexural strength, and TEC, of sintered mullite–cordierite composite pellets with a weight ratio of 70:30, at various sintering temperatures between 1300 °C and 1450 °C. As shown in Figure 6a, the density and porosity of mullite–cordierite composite pellets are typically inverse. At 1350 °C, the density is greater than at 1300 °C. When the sintering temperature is increased, some cordierite transforms into a glassy phase, which fills the porosity of the mullite–cordierite composites [5,23,26]. However, above 1400 °C, all cordierite melts into a glassy phase. The mismatch of TEC between mullite’s crystalline phase and cordierite’s glassy phase results in a loss of density, which results in the formation of large pores and macrocracks. In addition, as shown in Figure 6b, samples sintered at 1400 °C and 1450 °C exhibit a significant increase in TEC. This result can also be attributed to the thermal stress caused by the TEC mismatch between the glassy and crystalline phases of mullite [27]. The TEC of the sintered samples sintered at 1300 °C and 1350 °C, however, was 3.45 ppm/°C and 3.42 ppm/°C, respectively, which is comparable to the TEC of Si. This result explains why the sample sintered at 1350 °C is more resistant to thermal stresses than samples sintered at other temperatures. The flexural strength of the material can be explained using the density values shown in Figure 6a. The pellet made of mullite and cordierite sintered at 1350 °C has the most compact structure, which enhances its mechanical properties. Flexural strength is confirmed to be greatest at 1350 °C (264 MPa). The composite pellets sintered at other temperatures except 1350 °C, by contrast, show low flexural strength due to their relatively low density.

### 3.4. The Reliability and Durability of the Large-Area Mullite–Cordierite Composite Substrate

On the basis of the foregoing results, a large-area mullite–cordierite composite substrate was fabricated by sintering at 1350 °C with granules composed of 70 wt% mullite–30 wt% cordierite. The diameter of the large-area ceramic composite substrate is 320 mm. The thickness of the substrate is approximately 5.7 µm, and its relative standard deviation is 0.07%, indicating that it is highly uniform in thickness, as shown in Figure 7a. To create the probe card’s fine pitch, 30,000 micro-holes were drilled into the large-area substrate using ultrasonic drilling. The size and position deviations of the holes were determined in order to determine the shrinkage rate of the large-area substrate during the sintering process, as shown in Figure 7b,c. The SEM images in Figure 7b show the cross-sections of the ceramic substrate with the processed micro-holes. The average sizes of the processed holes’ top, middle, and bottom sections were 416.3 µm, 400.1 µm, and 387.8 µm, respectively, with a <5% deviation. Figure 7c shows enlarged SEM images of the four sections denoted by the numbers #1 to #4 in Figure 7a. In each of the SEM images, the distance between the holes was measured four times, and the standard deviation of the distance was determined to be <3%. These results indicate that the processed holes on the sintered large-area ceramic substrate are constant in size and the distance between the holes is also uniform. Additionally, it is expected that the hole size and the distance between the holes will hardly shrink during the electrical die sorting process, since the mullite–cordierite composite substrate was sintered at 1350 °C.

To assess the reliability and durability of the large-area ceramic composite substrate, it was cut into 80 mm × 80 mm specimens and subjected to environmental tests such as the DH and TS tests. Figure 8 shows the changes in the specimens’ sheet resistance during the environmental tests. In the DH test, the specimens were exposed to an 85% relative humidity at 85 °C. Resistance changed at a nearly constant rate. As can be seen, moisture absorption by the high temperature and humidity was minimal. In comparison to the DH test, a greater change in the specimen’s sheet resistance was observed after it was subjected to a TS test consisting of over 100 cycles at a temperature ranging from −40 °C to 85 °C. The impact of the thermal shock resulted in a change in sheet resistance. However, the resistance changed at a slow rate, indicating that the ceramic substrate was not significantly affected by abrupt temperature changes. In comparison to previous studies [5,10,15,25,28], this study conducted the TS test in a harsher environment at a lower temperature, though the temperature range used was slightly different. The fabricated specimens exhibited a low resistivity change of <10%, and no microcrack on the surface of the sample was observed even after 100 cycles. These findings demonstrate the superior reliability and durability of the mullite–cordierite substrate.

Figure 9 shows the effect of the environmental tests on the flexural strength of the ceramic substrate. Flexural strength was determined using specimens with dimensions of 3 mm × 4 mm × 36 mm, and the result was calculated as the average of ten specimen measurements. When the flexural strength of the specimens was compared before and after each environmental test, it was observed that the flexural strength of all the specimens decreased slightly following the environmental tests. It is reasonable to assume that thermal stress can result in the formation of an impurity, which has an effect on the physical properties [28]. In comparison to previously reported composite substrates [29,30], the fabricated large-area mullite–cordierite composite substrate exhibits superior mechanical strength under harsher environmental conditions. This is because it has a high flexural strength of more than 200 MPa and a flexural strength change rate of less than 10%, after 100 cycles of all environmental tests.

As a result, large-area substrates fabricated with spherical 70 wt% mullite-30 wt% cordierite granules with a homogeneous particle size distribution remained unaffected by environmental tests designed to accelerate fatigue failure. These findings demonstrate the composite’s high reliability and durability.

## 4. Conclusions

Sintered mullite–cordierite composite pellets were made from spherical composite granules prepared by spray drying fine mixed powders obtained through attrition milling. The composition ratio and sintering temperature of the composite pellets were varied to obtain pellets with excellent physical and mechanical properties. In particular, the mullite–cordierite composite pellet containing 70 wt% mullite and 30 wt% cordierite sintered at 1350 °C showed the lowest thermal expansion and the highest flexural strength. On the basis of the results, a large-area mullite–cordierite composite substrate with a diameter of 320 mm and a thickness of 5.7 µm was successfully fabricated. Due to the fabricated substrate’s low TEC, the processed holes on the large-area ceramic substrate shrank very little during the sintering process. Additionally, the rate of change in sheet resistance and flexural strength in the fabricated ceramic substrate was confirmed to be within 10% after exposure to harsh environmental tests, such as the DH test and TS test, indicating high reliability and durability. Thus, the mullite–cordierite composite material may be considered a suitable substrate for the probe card.

## Figures and Tables

**Figure 1 materials-15-04283-f001:**
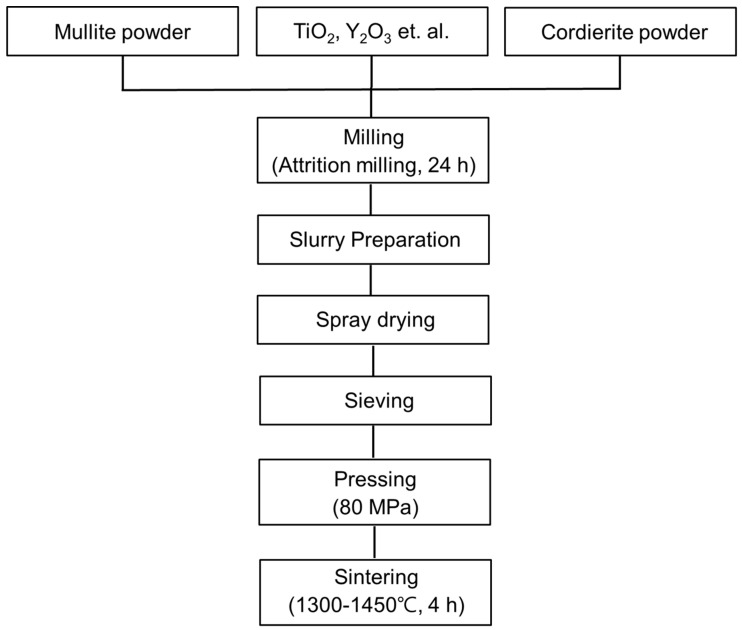
Flow diagram of the experimental procedure.

**Figure 2 materials-15-04283-f002:**
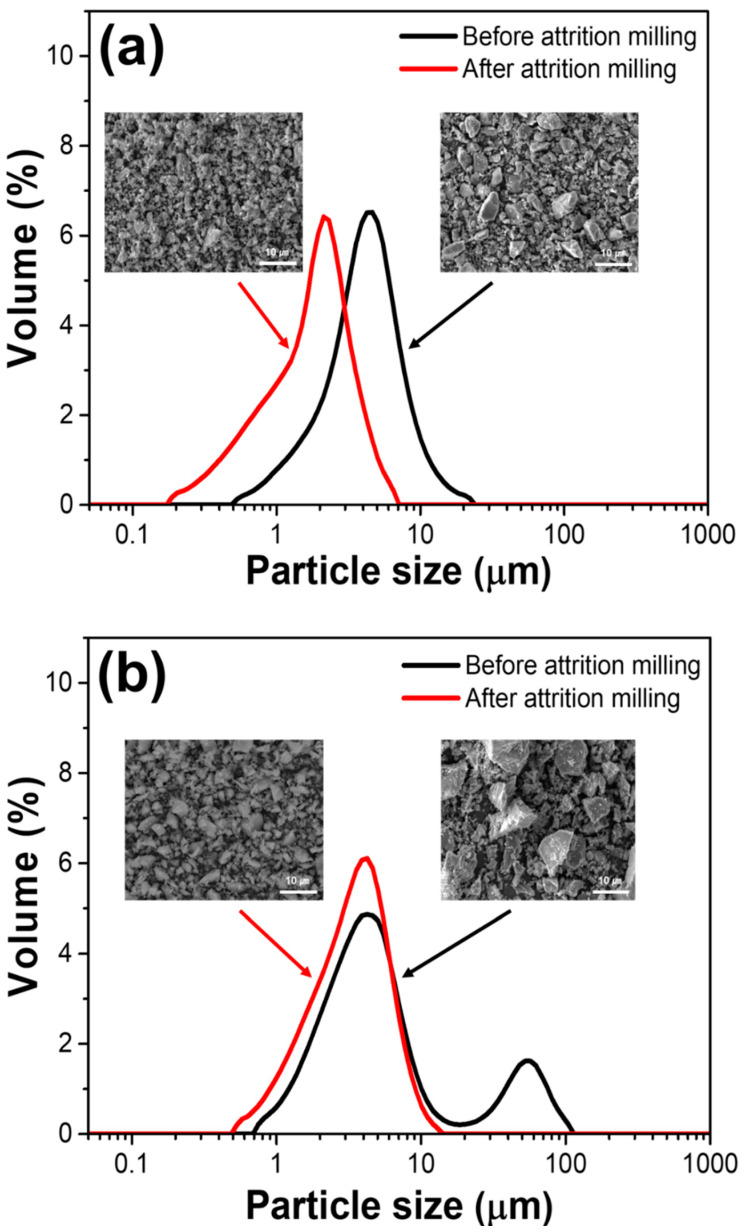
Particle size distributions of the starting powders: (**a**) mullite and (**b**) cordierite. The inset images show the SEM micrographs before and after attrition milling of the starting powders (mullite, cordierite).

**Figure 3 materials-15-04283-f003:**
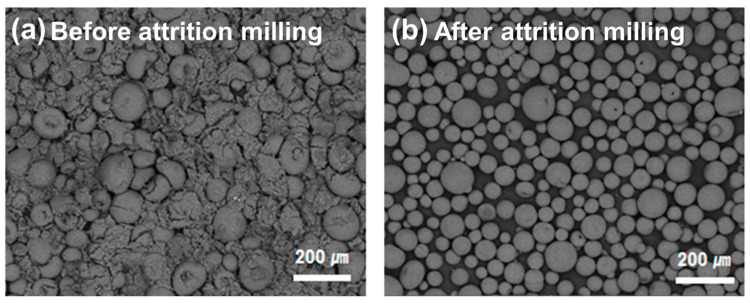
SEM images of the mullite–cordierite composite granules: (**a**) before and (**b**) after attrition milling.

**Figure 4 materials-15-04283-f004:**
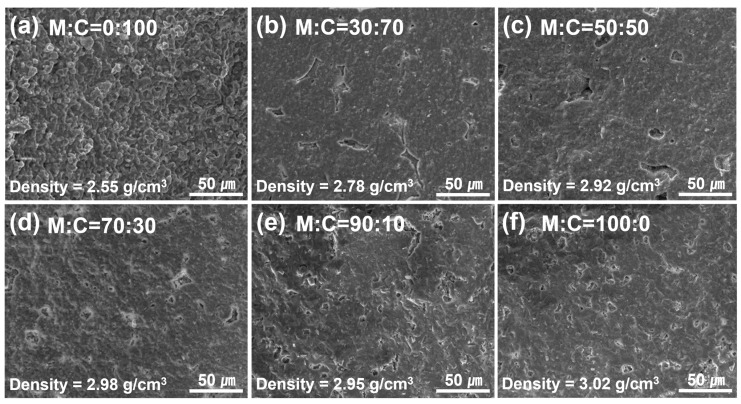
SEM images of the mullite–cordierite composite sintered at 1300 °C: (**a**) M:C = 0:100, (**b**) M:C = 30:70, (**c**) M:C = 50:50, (**d**) M:C = 70:30, (**e**) M:C = 90:10, and (**f**) M:C = 100:0.

**Figure 5 materials-15-04283-f005:**
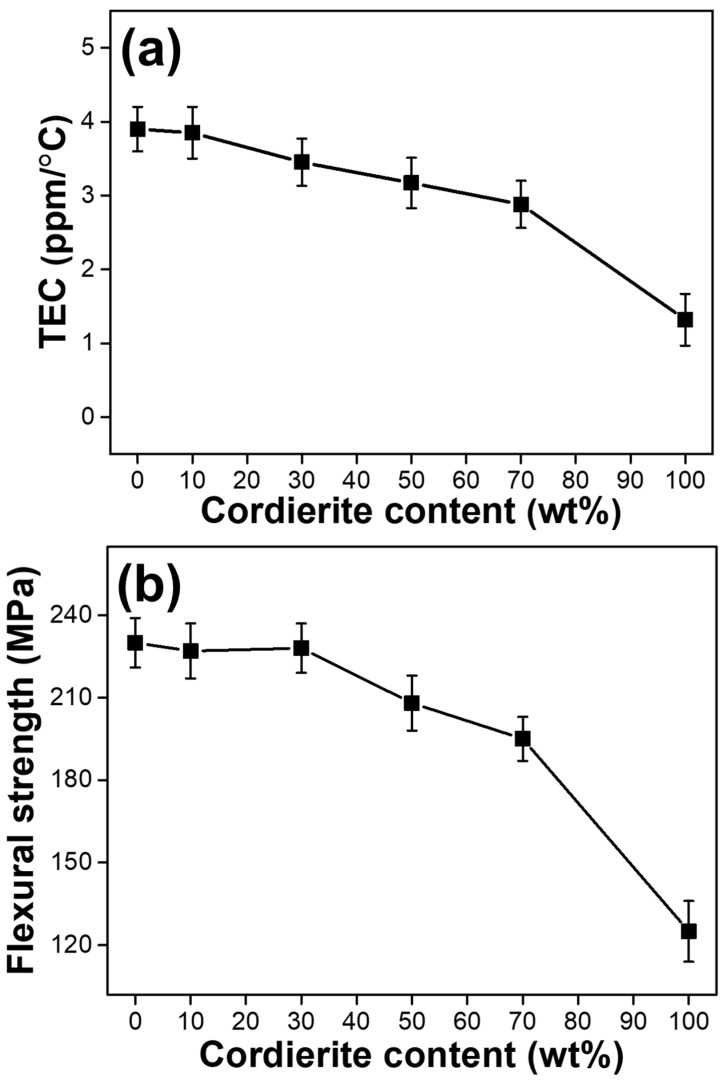
Variation in the physical and mechanical properties of mullite–cordierite composites with the cordierite content: (**a**) TEC and (**b**) flexural strength.

**Figure 6 materials-15-04283-f006:**
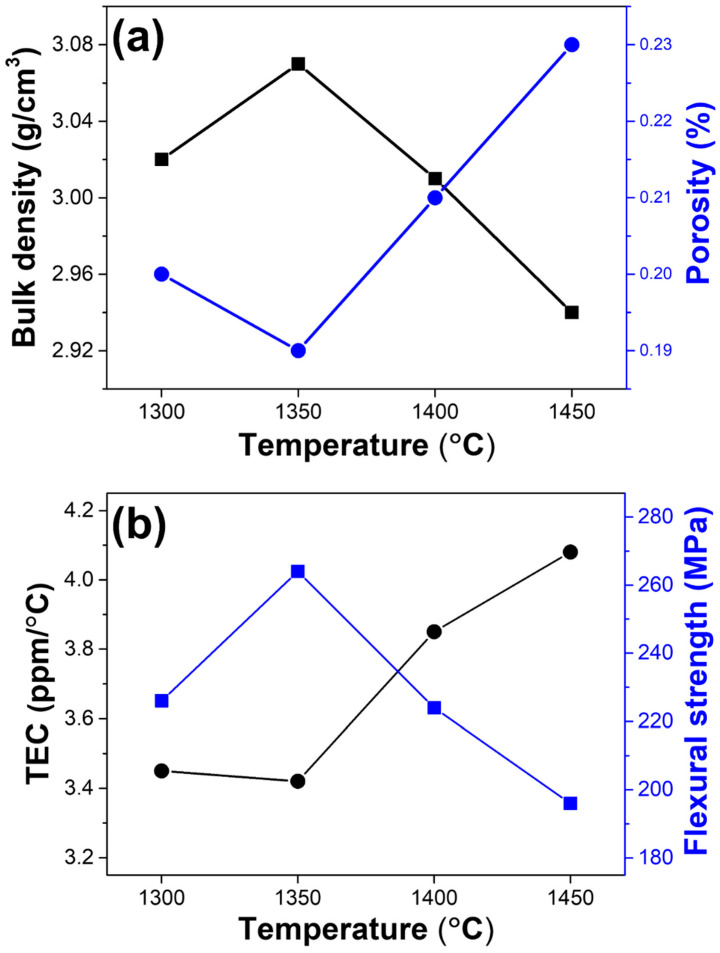
Variation in the physical and mechanical properties of the mullite–cordierite composites at different sintering temperatures: (**a**) bulk density and porosity and (**b**) TEC and flexural strength.

**Figure 7 materials-15-04283-f007:**
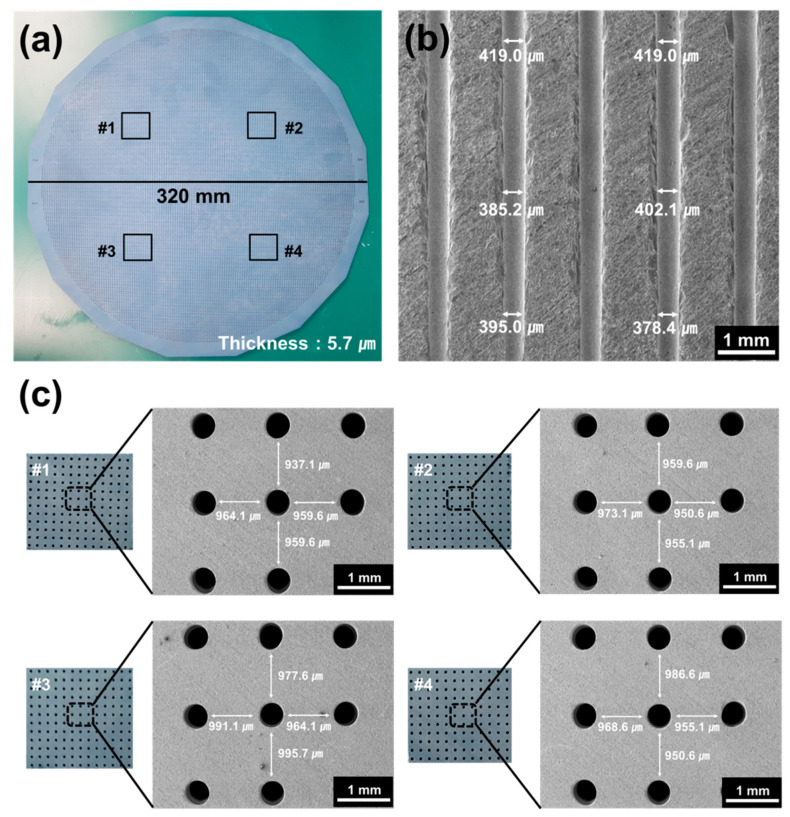
The ceramic substrate fabricated by using mullite–cordierite composite containing 70 wt% mullite–30 wt% cordierite sintered at 1350 °C: (**a**) large-area ceramic composite substrate, (**b**) cross-section of the ceramic composite substrate, and (**c**) enlarged SEM images of the four sections in (**a**).

**Figure 8 materials-15-04283-f008:**
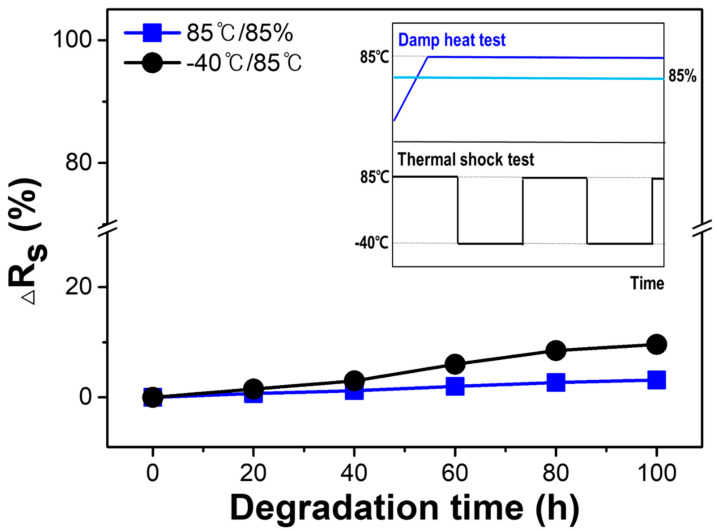
Variation in the sheet resistance of ceramic substrate under damp heat test (85 °C/85%) and thermal shock test (between −40 and 85 °C). The inset graph shows a schematic of the harsh environmental tests to which the ceramic substrate is subjected.

**Figure 9 materials-15-04283-f009:**
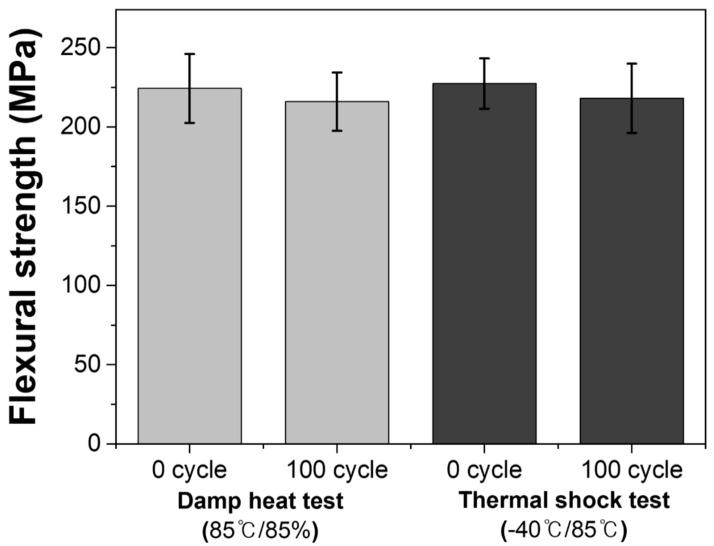
Variation of the flexural strength with the degradation time under damp heat test (85 °C/85%) and thermal shock test (between −40 and 85 °C).

## Data Availability

The data presented in this study are available on request from the corresponding author.

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
