# Peer review of "Fabrication of Large-Area Mullite–Cordierite Composite Substrates for Semiconductor Probe Cards and Enhancement of Their Reliability"

_materials, 2022, doi:10.3390/ma15124283_

Round 1

Reviewer 1 Report

Comments:

1. Please add parametric details of Ultrasonic drilling process (frequency, amplitude of the ultrasonic generator, the size of the processing pin, and the processing pressure) in section 2.2

2. Does Fig. 4 presents the fractured surfaces of mullite-cordierite composite pellets? Were these surfaces polished before imaging? Low magnification image would shed more light on the structural integrity of the pellets.

3. Please explain the decrease in density of sample in fig 4-e which subsequently increase for sample in fig 4-f.

4. Please explain the statement “These findings indicate that the processed holes on the large-area ceramic substrate are uniform in size and that they shrank very little during the sintering process” in section 3.4, paragraph 1. This statement cannot be true as the holes were processed on already sintered substrate.

Author Response

Dear Reviewer,

First of all,
I would like to thank you very much for your sincere comments and hope to meet the comments.
I also think it was a great opportunity to improve this article.

I wrote each answer to the comments you had pointed out in this file (Response to Reviewers). 
And, in the revised manuscript, I inserted the corresponding content and figures marked in red.

I am looking forward to your reply.
Thank you again.

Yours sincerely,

Reviewer 2 Report

Reviewed paper deals with the fabrication of mullite-cordierite composites and their properties. The Paper is clearly and comprehensibly written but some uncertainties were found.
The comments are listed below:
Section Experimental part:
The particle size measurement description is missing. What type of way was used- wet or dry, etc.?
The sintering conditions should be more specified. Please complete it.
The information about the number of tested samples and applied statistics is missing, especially in mechanical testing.
There is no information about the sample preparation for SEM analysis. Please complete it.

Section Results and Discussion:
The sintering process is not precisely described and should be completed. The discussion about the glassy phase presence is given only from literature data. The presented SEM pictures are not sufficient to support the proposed conclusions. The grain boundaries from the provided SEM pictures are not clearly visible. Polished and etched samples should be more appropriate to discuss the influence of composition. The influence of sintering aids is missing at all. The heating microscopy or similar analysis can provide more information about the sinter-crystallization process occurring around 1300-1400°C.
The more detailed SEM pictures and heating microscopy analysis will be beneficial to describe the sinter-crystallization process.
TEC, durability, and reliability data of sintering aid influence is missing. Please provide it.

Author Response

(The authors gave the same response as above.)
